# Precision and Recall for Time Series

**Nesime Tatbul** *
Intel Labs and MIT
tatbul@csail.mit.edu

**Tae Jun Lee** *
Microsoft
tae_jun_lee@alumni.brown.edu

**Stan Zdonik**
Brown University
sbz@cs.brown.edu

**Mejbah Alam**
Intel Labs
mejbah.alam@intel.com

**Justin Gottschlich**
Intel Labs
justin.gottschlich@intel.com

## Abstract

Classical anomaly detection is principally concerned with *point-based anomalies*, those anomalies that occur at a single point in time. Yet, many real-world anomalies are *range-based*, meaning they occur over a period of time. Motivated by this observation, we present a new mathematical model to evaluate the accuracy of time series classification algorithms. Our model expands the well-known *Precision* and *Recall* metrics to measure ranges, while simultaneously enabling customization support for domain-specific preferences.

## 1 Introduction

*Anomaly detection* (AD) is the process of identifying non-conforming items, events, or behaviors [1, 9]. The proper identification of anomalies can be critical for many domains. Examples include early diagnosis of medical diseases [22], threat detection for cyber-attacks [3, 18, 36], or safety analysis for self-driving cars [38]. Many real-world anomalies can be detected in time series data. Therefore, systems that detect anomalies should reason about them as they occur over a period of time. We call such events *range-based anomalies*. Range-based anomalies constitute a subset of both contextual and collective anomalies [9]. More precisely, a *range-based anomaly* is one that occurs over a consecutive sequence of time points, where no non-anomalous data points exist between the beginning and the end of the anomaly. The standard metrics for evaluating time series classification algorithms today, *Precision* and *Recall*, have been around since the 1950s. Originally formulated to evaluate document retrieval algorithms by counting the number of documents that were correctly returned against those that were not [6], *Precision* and *Recall* can be formally defined as follows [1] (where $TP, FP, FN$ are the number of true positives, false positives, false negatives, respectively):

$$Precision = TP \div (TP + FP) \tag{1}$$

$$Recall = TP \div (TP + FN) \tag{2}$$

Informally, *Precision* is the fraction of all detected anomalies that are real anomalies, whereas, *Recall* is the fraction of all real anomalies that are successfully detected. In this sense, *Precision* and *Recall* are complementary, and this characterization proves useful when they are combined (e.g., using $F_\beta$-*Score*, where $\beta$ represents the relative importance of *Recall* to *Precision*) [6]. Such combinations help gauge the quality of anomaly predictions. While useful for point-based anomalies, classical *Precision* and *Recall* suffer from the inability to represent domain-specific time series anomalies. This has a negative side-effect on the advancement of AD systems. In particular, many time series AD systems' accuracy is being misrepresented, because point-based *Precision* and *Recall* are being used to measure their effectiveness for range-based anomalies. Moreover, the need to accurately identify time series anomalies is growing in importance due to the explosion of streaming and real-time systems [2, 5, 16, 19, 27, 28, 30, 31, 37, 40].

To address this need, we redefine *Precision* and *Recall* to encompass range-based anomalies. Unlike prior work [2, 25], our new mathematical definitions extend their classical counterparts, enabling

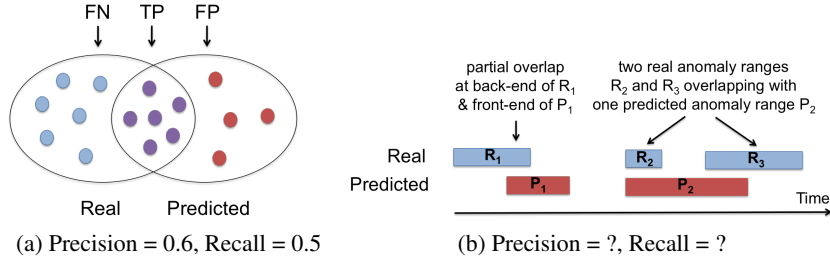

(a) Precision = 0.6, Recall = 0.5          (b) Precision = ?, Recall = ?

Figure 1: Point-based vs. range-based anomalies.

our model to subsume the classical one. Further, our formulation is more broadly generalizable by providing specialization functions that can control a domain's bias along a multi-dimensional axis to properly accommodate the needs of that specific domain. Thus, the key contribution of this paper is a new, customizable mathematical model, which can be used to evaluate and compare results of AD algorithms. Although outside the scope of this paper, our model can also be used as the objective function for machine learning (ML) training, which may have a profound impact on AD training strategies, giving rise to fundamentally new ML techniques in the future.

In the remaining sections, we first detail the problem and provide an overview of prior work. In Section 4, we formally present our new model. Section 5 provides a detailed experimental study of our new model compared to the classical model and a recent scoring model provided by Numenta [25]. Finally, we conclude with a brief discussion of future directions.

## 2   Problem motivation and design goals

Classical precision and recall are defined for sets of independent points (Figure 1a), which is sufficient for point-based anomalies. Time series AD algorithms, on the other hand, work with sequences of time intervals (Figure 1b) [11, 17, 18, 21, 23, 30]. Therefore, important time series specific characteristics cannot be captured by the classical model. Many of these distinctions are due to *partial overlaps*. In the point-based case, a predicted anomaly point is either a member of the set of real anomalies (a TP) or not (an FP). In the range-based case, a predicted anomaly range might partially overlap with a real one. In this case, a single prediction range is partially a TP and partially an FP at the same time. The size of this partial overlap needs to be quantified, but there may be additional criteria. For example, one may want to consider the *position of the overlap*. After all, a range consists of an ordered collection of points and the order might be meaningful for the application. For instance, detecting the earlier portion of an anomaly (i.e., its "front-end") might be more critical for a real-time application to reduce the time to react to it [24] (see Section 4.3 for more examples). Furthermore, overlaps are no longer "1-to-1" as in the classical model; one or more predicted anomaly ranges may (partially) overlap with one or more real ones. Figure 1b illustrates one such situation ($P_2$ vs. $R_2$, $R_3$). The specific domain might care if each independent anomaly range is detected as a single unit or not. Thus, we may also want to capture *cardinality* when measuring overlap.

If all anomalies were point-based, or if there were no partial-overlap situations, then the classical model would suffice. However, for general range-based anomalies, the classical model falls short. Motivated by these observations, we propose a new model with the following design goals:

- *Expressive*: captures criteria unique to range-based anomalies, e.g., overlap position and cardinality.
- *Flexible*: supports adjustable weights across multiple criteria for domain-specific needs.
- *Extensible*: supports inclusion of additional domain-specific criteria that cannot be known a priori.

Our model builds on the classical precision/recall model as a solid foundation, and extends it to be used more effectively for range-based AD (and time series classification, in general).

## 3   Related work

There is a growing body of research emphasizing the importance of time series classification [10, 19, 27], including anomaly detection. Many new ML techniques have been proposed to handle time series anomalies for a wide range of application domains, from space shuttles to web services [16, 30, 31, 41]. MacroBase [7] and SPIRIT [33] dynamically detect changes in time series when analyzing fast, streaming data. Lipton et al. investigated the viability of existing ML-based AD techniques on medical time series [29], while Patcha and Park have drawn out weaknesses for AD systems to

Table 1: Notation

| Notation | Description |
|---|---|
| $R, R_i$ | set of real anomaly ranges, the $i^{th}$ real anomaly range |
| $P, P_j$ | set of predicted anomaly ranges, the $j^{th}$ predicted anomaly range |
| $N, N_r, N_p$ | number of all points, number of real anomaly ranges, number of predicted anomaly ranges |
| $\alpha$ | relative weight of existence reward |
| $\gamma(), \omega(), \delta()$ | overlap cardinality function, overlap size function, positional bias function |

properly handle cyber-attacks in a time series setting, amongst others [34]. Despite these efforts, techniques designed to judge the efficacy of time series AD systems remain largely underdeveloped.

Evaluation measures have been investigated in other related areas [8, 13], e.g., sensor-based activity recognition [12], time series change point detection [4]. Most notably, Ward et al. provide a fine-grained analysis of errors by dividing activity events into segments to account for situations such as event fragmentation/merging and timing offsets [39]. Because these approaches do not take positional bias into account and do not provide a tunable model like ours, we see them as complementary.

Similar to our work, Lavin and Ahmad have also noted the lack of proper time series measurement techniques for AD in their Numenta Anomaly Benchmark (NAB) work [25]. Yet, unlike ours, the NAB scoring model remains point-based in nature and has a fixed bias (early detection), which makes it less generalizable than our model. We rediscuss NAB in greater detail in Section 5.3.

## 4 Precision and recall for ranges

In this section, we propose a new way to compute precision and recall for ranges. We first formulate recall, and then follow a similar methodology to formulate precision. Finally, we provide examples and guidelines to illustrate the practical use of our model. Table 1 summarizes our notation.

### 4.1 Range-based recall

The basic purpose of recall is to reward a prediction system when real anomalies are successfully identified (TPs), and to penalize it when they are not (FNs). Given a set of real anomaly ranges $R = \{R_1, .., R_{N_r}\}$ and a set of predicted anomaly ranges $P = \{P_1, .., P_{N_p}\}$, our $Recall_T(R, P)$ formulation iterates over the set of all real anomaly ranges ($R$), computing a recall score for each real anomaly range ($R_i \in R$) and adding them up into a total recall score. This total score is then divided by the total number of real anomalies ($N_r$) to obtain an average recall score for the whole time series.

$$Recall_T(R, P) = \frac{\sum_{i=1}^{N_r} Recall_T(R_i, P)}{N_r} \tag{3}$$

When computing the recall score $Recall_T(R_i, P)$ for a single anomaly range $R_i$, we consider:

- *Existence*: Catching the existence of an anomaly (even by predicting only a single point in $R_i$), by itself, might be valuable for the application.
- *Size*: The larger the size of the correctly predicted portion of $R_i$, the higher the recall score.
- *Position*: In some cases, not only size, but also the relative position of the correctly predicted portion of $R_i$ might matter to the application.
- *Cardinality*: Detecting $R_i$ with a single prediction range $P_j \in P$ may be more valuable than doing so with multiple different ranges in $P$ in a fragmented manner.

We capture all of these considerations as a sum of two main reward terms weighted by $\alpha$ and $(1 - \alpha)$, respectively, where $0 \leq \alpha \leq 1$. $\alpha$ represents the relative importance of rewarding *existence*, whereas $(1 - \alpha)$ represents the relative importance of rewarding *size*, *position*, and *cardinality*, all of which stem from the actual overlap between $R_i$ and the set of all predicted anomaly ranges ($P_j \in P$).

$$Recall_T(R_i, P) = \alpha \times ExistenceReward(R_i, P) + (1 - \alpha) \times OverlapReward(R_i, P) \tag{4}$$

If anomaly range $R_i$ is identified (i.e., $|R_i \cap P_j| \geq 1$ across all $P_j \in P$), then a reward is earned.

$$ExistenceReward(R_i, P) = \begin{cases} 1, \text{if } \sum_{j=1}^{N_p} |R_i \cap P_j| \geq 1 \\ 0, \text{otherwise} \end{cases} \tag{5}$$

Additionally, a cumulative overlap reward that depends on three functions, $0 \leq \gamma() \leq 1, 0 \leq \omega() \leq 1$, and $\delta() \geq 1$, is earned. These capture the *cardinality* ($\gamma()$), *size* ($\omega()$), and *position* ($\delta()$) aspects of

```
function ω(AnomalyRange, OverlapSet, δ)
    MyValue ← 0
    MaxValue ← 0
    AnomalyLength ← length(AnomalyRange)
    for i ← 1, AnomalyLength do
        Bias ← δ(i, AnomalyLength)
        MaxValue ← MaxValue + Bias
        if AnomalyRange[i] in OverlapSet then
            MyValue ← MyValue + Bias
    return MyValue/MaxValue
```

(a) Overlap size

```
function δ(i, AnomalyLength)          ▷ Flat bias
    return 1
function δ(i, AnomalyLength) ▷ Front-end bias
    return AnomalyLength - i + 1
function δ(i, AnomalyLength) ▷ Back-end bias
    return i
function δ(i, AnomalyLength)       ▷ Middle bias
    if i ≤ AnomalyLength/2 then
        return i
    else
        return AnomalyLength - i + 1
```

(b) Positional bias

Figure 2: Example definitions for $\omega()$ and $\delta()$ functions.

the overlap. More specifically, the cardinality term serves as a scaling factor for the rewards earned from overlap size and position.

$$OverlapReward(R_i, P) = CardinalityFactor(R_i, P) \times \sum_{j=1}^{N_p} \omega(R_i, R_i \cap P_j, \delta) \qquad (6)$$

When $R_i$ overlaps with only one predicted anomaly range, the cardinality factor reward is the largest (i.e., 1). Otherwise, it receives $0 \leq \gamma() \leq 1$ defined by the application.

$$CardinalityFactor(R_i, P) = \begin{cases} 1 & \text{, if } R_i \text{ overlaps with at most one } P_j \in P \\ \gamma(R_i, P), \text{otherwise} \end{cases} \qquad (7)$$

Note that both the weight ($\alpha$) and the functions ($\gamma()$, $\omega()$, and $\delta()$) are tunable according to the needs of the application. We illustrate how they can be customized with examples in Section 4.3.

## 4.2 Range-based precision

As seen in Equations 1 and 2, the key difference between precision and recall is that precision penalizes FPs instead of FNs. Given a set of real anomaly ranges $R = \{R_1, .., R_{N_r}\}$ and a set of predicted anomaly ranges $P = \{P_1, .., P_{N_p}\}$, our $Precision_T(R, P)$ formula iterates over the set of all predicted anomaly ranges ($P$), computing a precision score for each predicted anomaly range ($P_i \in P$) and adding them up into a total precision score. This total score is then divided by the total number of predicted anomalies ($N_p$) to obtain an average precision score for the whole time series.

$$Precision_T(R, P) = \frac{\sum_{i=1}^{N_p} Precision_T(R, P_i)}{N_p} \qquad (8)$$

When computing $Precision_T(R, P_i)$ for a single predicted anomaly range $P_i$, there is no need for an existence reward, since precision by definition emphasizes prediction quality, and existence by itself is too low a bar for judging the quality of a prediction (i.e., $\alpha = 0$). On the other hand, the overlap reward is still needed to capture the cardinality, size, and position aspects of a prediction.

$$Precision_T(R, P_i) = CardinalityFactor(P_i, R) \times \sum_{j=1}^{N_r} \omega(P_i, P_i \cap R_j, \delta) \qquad (9)$$

Like in our recall formulation, the $\gamma()$, $\omega()$, and $\delta()$ functions are tunable according to application semantics. Despite not explicitly shown in our notation, these functions can be defined differently for $Recall_T$ and $Precision_T$ (see Sections 4.3 and 5 for examples).

## 4.3 Customization guidelines and examples

Figure 2a provides an example for the $\omega()$ function for size, which can be used with a $\delta()$ function for positional bias. In general, for both $Recall_T$ and $Precision_T$, we expect $\omega()$ to be always defined like in our example, due to its additive nature and direct proportionality with the size of the overlap.

$\delta()$, on the other hand, can be defined in multiple different ways, as illustrated with four examples in Figure 2b. If all index positions of an anomaly range are equally important for the application, then the flat bias function should be used. In this case, simply, the larger the size of the overlap, the higher the overlap reward will be. If earlier (later) index positions carry more weight than later (earlier) ones, then the front-end (back-end) bias function should be used instead. Finally, the middle bias function is for prioritizing mid-portions of anomaly ranges. In general, we expect $\delta()$ to be a function in which the value of an index position $i$ monotonically increases/decreases based on its

distance from a well-defined reference point of the anomaly range. From a semantic point of view, $\delta()$ signifies the urgency of when an anomaly is detected and reacted to. For example, front-end bias is preferable in scenarios where early response is critical, such as cancer detection or real-time systems. Back-end bias is useful in scenarios where responses are irreversible (e.g., firing of a missile, output of a program). In such scenarios, it is generally more desirable to delay raising detection of an anomaly until there is absolute certainty of its presence. Middle bias can be useful when there is a clear trade-off between the effect of early and late detection, i.e., an anomaly should not be reacted to too early or too late. Hard braking in autonomous vehicles is an example: braking too early may unnecessarily startle passengers, while braking too late may cause a collision. We expect that for $Recall_T$, $\delta()$ will typically be set to one of the four functions in Figure 2b depending on the domain; whereas a flat bias function will suffice for $Precision_T$ in most domains, since an FP is typically considered uniformly bad wherever it appears in a prediction range.

We expect the $\gamma()$ function to be defined similarly for both $Recall_T$ and $Precision_T$. Intuitively, in both cases, the cardinality factor should be inversely proportional to the number of distinct ranges that a given anomaly range overlaps. Thus, we expect $\gamma()$ to be generally structured as a reciprocal rational function $1/f(x)$, where $f(x) \geq 1$ is a single-variable polynomial and $x$ represents the number of distinct overlap ranges. A typical example for $\gamma()$ is $1/x$.

Before we conclude this section, it is important to note that our precision and recall formulas subsume their classical counterparts, i.e., $Recall_T \equiv Recall$ and $Precision_T \equiv Precision$, when:

(i) all $R_i \in R$ and $P_j \in P$ are represented as unit-size ranges (e.g., range $[1, 3]$ as $[1, 1], [2, 2], [3, 3]$),
(ii) $\alpha = 0$, $\gamma() = 1$, $\omega()$ is as in Figure 2a, and $\delta()$ returns flat positional bias as in Figure 2b.

## 5   Experimental study

In this section, we present an experimental study of our range-based model applied to results of state-of-the-art AD algorithms over a collection of diverse time series datasets. We aim to demonstrate two key features of our model: *(i)* its expressive power for range-based anomalies compared to the classical model, *(ii)* its flexibility in supporting diverse scenarios in comparison to the Numenta scoring model [25]. Furthermore, we provide a cost analysis for computing our new metrics, which is important for their practical application.

### 5.1   Setup

**System:** All experiments were run on a Windows 10 machine with an Intel® Core™ i5-6300HQ processor running at 2.30 GHz with 4 cores and 8 GB of RAM.
**Datasets:** We used a mixture of real and synthetic datasets. The real datasets are taken from the NAB Data Corpus [32], whereas the synthetic datasets are generated by the Paranom tool [15]. All data is time-ordered, univariate, numeric time series, for which anomalous points/ranges (i.e., "the ground truth") are already known. *NYC-Taxi:* A real dataset collected by the NYC Taxi and Limousine Commission. The data represents the number of passengers over time, recorded as 30-minute aggregates. There are five anomalies: the NYC Marathon, Thanksgiving, Christmas, New Year's day, and a snow storm. *Twitter-AAPL:* A real dataset with a time series of the number of tweets that mention Apple's ticker symbol (AAPL), recorded as 5-minute aggregates. *Machine-Temp:* A real dataset of readings from a temperature sensor attached to a large industrial machine, with three anomalies: a planned shutdown, an unidentified error, and a catastrophic failure. *ECG:* A dataset from Paranom based on a real Electrocardiogram (ECG) dataset [20, 30]; augmented to include additional synthetic anomalies over the original single pre-ventricular contraction anomaly. *Space-Shuttle:* A dataset based on sensor measurements from valves on a NASA space shuttle; also generated by Paranom based on a real dataset from the literature with multiple additional synthetic anomalies [20, 30]. *Sine:* A dataset from Paranom that includes a sine wave oscillating between 0.2 and 0.5 with a complete period over 360 timestamps. It contains many stochastic anomalous events ranging from $50 - 100$ time intervals. *Time-Guided:* A Paranom dataset with monotonically increasing univariate values, in which multiple range-based stochastic anomalies appear with inverted negative values.
**Anomaly Detectors:** For our first set of experiments, we trained a TensorFlow-implemented version of LSTM-AD, a long short-term memory model for AD [30], for all datasets. For training and testing, we carefully partition each dataset to ensure anomaly ranges are intact in spite of the segmentation. We interpret adjacent anomalous points as part of a single anomaly range. Thus, the LSTM-AD testing phase output is a sequence of predicted anomaly ranges for each dataset. Secondly, to illustrate how our model can be used in evaluating and comparing predictions from multiple detectors, we repeat this procedure with two additional detectors: Greenhouse [26] and Luminol [28].

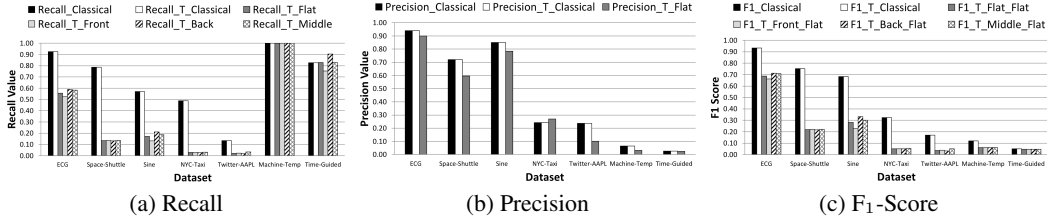

(a) Recall                                       (b) Precision                                    (c) F$_1$-Score

Figure 3: Our model vs. the classical point-based model.

**Compared Scoring Models:** We evaluated the prediction accuracy of each output from the anomaly detectors using three models: classical point-based precision/recall, the Numenta scoring model [25] (described in detail below), and our range-based precision/recall model. Unless otherwise stated, we use the following default parameter settings for computing our $Recall_T$ and $Precision_T$ equations: $\alpha = 0, \gamma() = 1, \omega()$ is as in Figure 2a, and $\delta()$ returns flat bias as in Figure 2b.

### 5.2 Comparison to the classical point-based model

First, we compare our range-based model with the classical point-based model. The goal of this experiment is twofold: *(i)* verify that our model subsumes the classical one, *(ii)* show that our model can capture additional variations for anomaly ranges that the classical one cannot.

We present our results in Figure 3 with three bar charts, which show Recall, Precision, and F$_1$-Score values for our LSTM-AD testing over the seven datasets, respectively. Recall values are computed using Equation 2 for $Recall$ (bar labeled `Recall_Classical`) and Equation 3 for $Recall_T$ (bars labeled `Recall_T_*`). Similarly, Precision values are computed using Equation 1 for $Precision$ (bar labeled `Precision_Classical`) and Equation 8 for $Precision_T$ (bars labeled `Precision_T_*`). Finally, *F$_1$-Score* values are computed using the following well-known equation, which essentially represents the harmonic mean of *Precision* and *Recall* [6]:

$$F_1\text{-}Score = 2 \times \frac{Precision \times Recall}{Precision + Recall} \tag{10}$$

We first observe that, in all graphs, the first two bars are equal for all datasets. *This demonstrates that our model, when properly configured as described at the end of Section 4.3, subsumes the classical one*. Next, we analyze the impact of positional bias ($\delta()$) on each metric. Note that $\gamma() = 1/x$ in what follows.

**Recall:** In Figure 3a, we provide $Recall_T$ measurements for four positional biases: flat, front, back, and middle (see Figure 2b). Recall is perfect (= 1.0) in all runs for the *Machine-Temp* dataset. When we analyze the real and predicted anomaly ranges for this dataset, we see two real anomaly ranges that LSTM-AD predicts as a single anomaly range. Both real anomalies are fully predicted (i.e., no false negatives and $x = 1$ for both ranges). Therefore, recall has the largest possible value independent from what $\delta()$ is, and is the expected behavior for this scenario.

For all other datasets except *Time-Guided*, $Recall_T$ is smaller than $Recall$. This is generally expected, as real anomaly ranges are rarely captured *(i)* entirely by LSTM-AD (i.e., overlap reward is partially earned) and *(ii)* by exactly one predicted anomaly range (i.e., cardinality factor downscales the overlap reward). Analyzing the real and predicted anomalies for *Time-Guided* reveals that LSTM-AD predicts 8/12 ranges fully and 4/12 ranges half-way (i.e., no range is completely missed and $x = 1$ for all 12 ranges). Furthermore, differences among $Recall_T$ biases are mainly due to the four half-predicts. All half-predicts lie at back-ends, which explains why `Recall_T_Back` is larger than other biases. *This illustrates positional bias sensitivity of $Recall_T$, which cannot be captured by the classical $Recall$.*

We make similar observations for *ECG* and *Sine*. That is, different positional bias leads to visibly different $Recall_T$ values. These demonstrate the sensitivity of $Recall_T$ to the positions of correctly predicted portions of anomalies. For *Space-Shuttle*, *NYC-Taxi*, and *Twitter-AAPL*, the differences are not as pronounced. Data indicates that these datasets contains few real anomaly ranges (5, 3, and 2, respectively), but LSTM-AD predicts a significantly larger number of small anomaly ranges. As such, overlaps are small and highly fragmented, likely making $\delta()$ less dominant than $\omega()$ and $\gamma()$.

**Precision:** In Figure 3b, we present $Precision_T$ only with flat positional bias, as we expect this to be the typical use case (see Section 4.3). In general, we observe that $Precision$ and $Precision_T$ follow a similar pattern. Also, $Precision_T$ is typically smaller than $Precision$ in all but two datasets (*Time-Guided* and *NYC-Taxi*). For *Time-Guided*, precision values turn out to be almost identical,

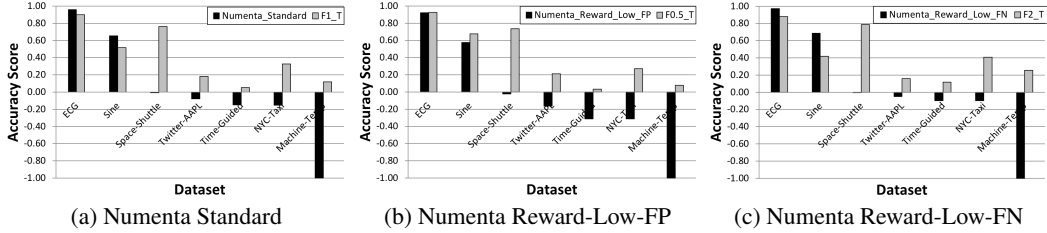

| (a) Numenta Standard | (b) Numenta Reward-Low-FP | (c) Numenta Reward-Low-FN |

Figure 4: Our model vs. the Numenta model.

because both real and predicted anomaly ranges are too narrow, diminishing the differences between points and ranges. When reviewing the result for *NYC-Taxi*, we found many narrow-range predictions against 3 wide-range real anomalies, with a majority of the predictions being false positives. Because the narrow predictions earn similar overlap rewards as unit-size/point predictions, yet the number of predicted anomaly ranges ($N_p$) is relatively smaller than the total number of points in those $N_p$ ranges, the value of $Precision_T$ comes out slightly larger than $Precision$ for this dataset. *Overall, this scenario demonstrates that $Precision_T$ is more range-aware than $Precision$ and can, therefore, judge exactness of range-based predictions more accurately.*

**$F_1$-Score:** In Figure 3c, we present $F_1$-Scores. The general behavior is as in the classical model, because $F_1$ is the harmonic mean of respective recall and precision values in both models. A noteworthy observation is that the $F_1$-Scores display similar sensitivity to positional bias variations as in recall graph of Figure 3a. *This demonstrates that our range-based model's expressiveness for recall and precision carries into combined metrics like $F_1$-Score.*

### 5.3 Comparison to the Numenta Anomaly Benchmark (NAB) scoring model

In this section, we report results from our LSTM-AD experiments with the Numenta Anomaly Benchmark (NAB) scoring model [25]. Our goals are: *(i)* to determine if our model can mimic the NAB model and *(ii)* to examine some of the flexibilities our model has beyond the NAB model.

**Background on NAB's Scoring Model:** While focusing entirely on real-time streaming applications, the NAB authors claim that: *(i)* only early anomaly detection matters (i.e., what we call front-end bias) and *(ii)* only the first detection point of an anomaly range matters (i.e., an emphasis on precision over recall and a bias toward single-point TP prediction systems). They then propose a scoring model, based on anomaly windows, application profiles, and a sigmoidal scoring function, which is specifically designed to reward detections earlier in a window. NAB's scoring model also includes weights that can be adjusted according to three predefined application profiles: Standard, Reward-Low-FP, and Reward-Low-FN. While this approach may be suitable for restricted domains, it does not generalize to all range-based anomalies. For example, in the identification of medical anomalies (e.g., cancer detection), it is critical to identify when an illness is regressing due to the use of life-threatening medical treatments [14]. Under NAB's single-point reward system, there is no clear distinction for such state changes. Furthermore, as discussed in a follow-up analysis by Singh and Olinsky [35], the NAB scoring system has a number of limitations that make it challenging to use in real-world applications, even within its restricted domain (e.g., determining anomaly window size a priori, ambiguities in scoring function, magic numbers, non-normalized scoring due to no lower bound). Given these irregularities, it is difficult, and perhaps ill-advised, to make a direct and unbiased comparison to the NAB scoring model. Instead, we focus on comparing relative approximations of our $F_\beta$-Scores compared to NAB scores, rather than their absolute values.

To obtain a behavior similar to NAB, we used the following settings for our model: $\alpha = 0$, $\gamma() = 1$, $\omega()$ is as in Figure 2a, $\delta()$ is front bias for $Recall_T$ and flat bias for $Precision_T$. Because NAB's scoring model makes point-based predictions, we represent $P_i$ as points. For each run, we compute `Recall_T_Front` and `Precision_T_Flat` and their $F_\beta$-Score corresponding to the NAB application profile under consideration (i.e., `F1_T` for `Numenta_Standard`, `F0.5_T` for `Numenta_Reward_Low_FP`, and `F2_T` for `Numenta_Reward_Low_FN`).

Figure 4 contains three bar charts, one for each Numenta profile. The datasets on the x-axis are presented in descending order of the Numenta accuracy scores to facilitate comparative analysis. For ease of exposition, we scaled the negative Numenta scores by a factor of 100 and lower-bounded the y-axis at -1. At a high level, although the accuracy values vary slightly across the charts, their approximation follows a similar pattern. Therefore, we only discuss Figure 4a.

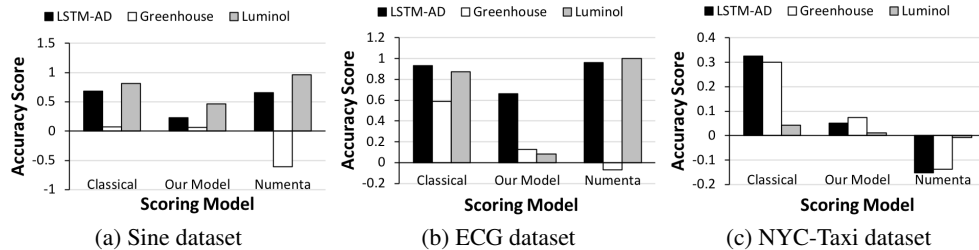

(a) Sine dataset       (b) ECG dataset       (c) NYC-Taxi dataset

Figure 5: Evaluating multiple anomaly detectors.

In Figure 4a, both models generally decrease in value from left to right except for a few exceptions, the *Space-Shuttle* dataset being the most striking. In analyzing the data for the *Space-Shuttle*, we noted a small number of mid-range real anomalies and a larger number of mid-range predicted anomalies. On the other hand, when we analyzed similar data for *Sine*, we found it had many narrow-range real and predicted anomalies that appear close together. These findings reveal that *Space-Shuttle* data had more FPs and TPs, which is reflected in its smaller $Precision_T$ score and larger $Recall_T$ score (Figure 3). Because NAB favors precision over recall (i.e., NAB heavily penalizes FPs), this discrepancy is further magnified. *Overall, this shows that not only can our model mimic the NAB scoring system, but it can also identify additional intricacies missed by NAB.*

Table 2: Sensitivity to positional bias

|                | Numenta_Standard | F1_T_Front_Flat | F1_T_Back_Flat |
|----------------|------------------|-----------------|----------------|
| Front-Predicted | 0.67            | 0.42            | 0.11           |
| Back-Predicted  | 0.63            | 0.11            | 0.42           |

To further illustrate, we investigated how the two models behave under two contrasting positional bias scenarios: *(i)* anomaly predictions overlapping with front-end portions of real anomaly ranges (*Front-Predicted*) and *(ii)* anomaly predictions overlapping with back-end portions of real anomaly ranges (*Back-Predicted*) as shown in Table 2. We artificially generated the desired ranges in a symmetric fashion to make them directly comparable. We then scored them using $Numenta\_Standard$, $F1\_T\_Front\_Flat$, and $F1\_T\_Back\_Flat$. As shown in Table 2, NAB's scoring function is not sufficiently sensitive to distinguish between the two scenarios. This is not surprising, as NAB was designed to reward early detections. However, our model distinguishes between the two scenarios when its positional bias is set appropriately. *This experiment demonstrates our model's flexibility and generality over the NAB scoring model.*

## 5.4 Evaluating multiple anomaly detectors

We analyzed the three scoring models for their effectiveness in evaluating and comparing prediction results from multiple anomaly detectors: LSTM-AD [30], Greenhouse [26], and Luminol (bitmap) [28]. Please note that the primary goal of this experiment is to compare the scoring models' behaviors for a specific application scenario, not to evaluate the AD algorithms themselves. In this experiment, we suppose an application that requires early detection of anomaly ranges in a non-fragmented manner, with precision and recall being equally important. In our model, this corresponds to the following settings: $\alpha = 0$, $\gamma() = 1/x$ for both $Precision_T$ and $Recall_T$, $\delta()$ is front bias for $Recall_T$, $\delta()$ is flat bias for $Precision_T$, and $\beta = 1$ for $F_\beta$-Score. In Numenta, the closest application profile is Standard, whereas in the classical model, the only tunable parameter is $\beta = 1$ for $F_\beta$-Score.

Figure 5 presents our results from running the detectors on three selected datasets: *Sine* (synthetic), *ECG* (augmented), and *NYC-Taxi* (real). In Figure 5a for *Sine*, all scoring models are in full agreement that Luminol is the most accurate and Greenhouse is the least. Detailed analysis of the data reveals that Luminol scores by far the greatest in precision, whereas Greenhouse scores by far the smallest. All scoring models recognize this obvious behavior equally well. In Figure 5b for *ECG*, all scoring models generally agree that Greenhouse performs poorly. It turns out that FPs clearly dominate in Greenhouse's anomaly predictions. Classical and Numenta score LSTM-AD and Luminol similarly, while our model strongly favors LSTM-AD over the others. Data indicates that this is because LSTM-AD makes a relatively smaller number of predictions clustered around the real anomaly ranges, boosting the number of TPs and making both precision and recall relatively greater. Luminol differs from LSTM-AD mainly in that it makes many more predictions, causing more fragmentation and severe score degradation due to $\gamma()$ in our model. Finally, in Figure 5c for *NYC-Taxi*, all scoring models largely disagree on the winner, except that LSTM-AD and Greenhouse perform somewhat similarly according to all. Indeed, both of these detectors make many narrow predictions against

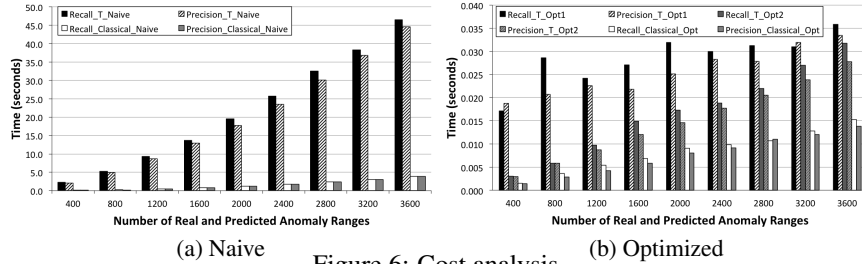

| (a) Naive | (b) Optimized |
|---|---|

Figure 6: Cost analysis.

3 relatively wide real anomaly ranges. Our model scores Greenhouse the best, mainly because its front-biased $Recall_T$ comes out to be much greater than the others'. Unlike the two precision/recall models, Numenta strongly favors Luminol, even though this detector misses one of the 3 real anomaly ranges in its entirety. *Overall, this experiment shows the effectiveness of our model in capturing application requirements; in particular, when comparing results of multiple anomaly detectors, and even in the presence of subtleties in data.*

## 5.5 Cost analysis

In this section, we analyze the computational overhead of our $Recall_T$ and $Precision_T$ equations.
**Naive:** In our $Recall_T$ and $Precision_T$ equations, we keep track of two sets, $R$ and $P$. A naive algorithm, which compares each $R_i \in R$ with all $P_j \in P$ for $Recall_T$ (and vice versa for $Precision_T$) has a computational complexity of $O(N_r \times N_p)$.
**Optimization1:** We can reduce the comparisons by taking advantage of sequential relationships among ranges. Assume that all ranges are represented as timestamp pairs, ordered by their first timestamps. We iterate over $R$ and $P$ simultaneously, only comparing those pairs that are in close proximity in terms of their timestamps. This optimized algorithm brings the computational complexity to $O(max\{N_r, N_p\})$.
**Optimization2:** An additional optimization is possible in computing the positional bias functions (e.g., flat). Instead of computing them for each point, we apply them in closed form, leading to a single computation per range. We see room for further optimizations (e.g., using bit vectors and bitwise operations), but exhaustive optimization is beyond the goals of our current analysis.

We implemented five alternative ways of computing recall and precision, and analyzed how the computation time for each of them scales with increasing number of real and predicted anomaly ranges. Two of these are naive algorithms for both classical and our recall/precision equations. The rest are their optimized variants. Ranges used in this experiment are randomly generated from a time series of 50K data points. Figure 6 shows our results. We plot naive and optimized approaches in separate graphs, as their costs are orders of magnitude different. In Figure 6a, we see that computing our range-based metrics naively is significantly more costly than doing so for the classical ones. Fortunately, we can optimize our model's computational cost. The optimized results in Figure 6b provide a clear improvement in performance. Both the classical metrics and ours can be computed in almost three orders of magnitude less time than their corresponding naive baselines when optimized. There is still a factor of 2-3 difference, but our model's values are notably closer at this scale to the classical model. *This analysis shows that our new range-based metrics can be computed efficiently, and their overhead compared to the classical ones is acceptably low.*

## 6 Conclusions and future directions

Precision and recall are commonly used to evaluate the accuracy of anomaly detection systems. Yet, classical precision and recall were designed for point-based data, whereas for time series data, anomalies are in the form of ranges. In response, we offer a new formal model that accounts for range-specific issues, such as partial overlaps between real vs. predicted ranges as well as their relative positions. Moreover, our approach allows users to supply customizable bias functions that enable weighing multiple criteria differently. We ran experiments with diverse datasets and anomaly detectors, comparing our model against two others, which illustrated that our new model is expressive, flexible, and extensible. Our evaluation model comes with a number of tunable parameters. Given an application, these parameters must properly set to suit the needs of that domain. Though our model is extensible, we expect that the example settings and guidelines we provided in the paper will be sufficient in most practical domains. In ongoing work, we have been actively collaborating with domain experts to apply our model in real use cases, such as autonomous driving. Future work includes designing new range-based ML techniques that are optimized for our new accuracy model.

**Acknowledgments**

We thank Eric Metcalf for his help with experiments. This research has been funded in part by Intel and by NSF grant IIS-1526639.

## Footnotes

* Lead authors.

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
