[Supplementary Material 1 · Paper968-supplementary-MultipleDetectors.pdf]

# NeurIPS'18 Paper #968:
# Precision and Recall for Time Series

## Supplementary Material
## for Section 5.4

# Readme

- This file contains line graphs for our 3 experimental datasets for LSTM-AD, Greenhouse, and Luminol.

- For each dataset, we plot real and predicted anomaly ranges for their respective time series.

- On each graph, x-axis represents time and y-axis denotes where anomalies are. (t,1) means that the data value at timestamp x=t is anomalous, whereas (t,0) means that it is not.

- We used these graphs as reference in understanding and explaining the experimental results in Section 5.4 of our paper.

Sine, LSTM-AD

**Real Anomaly Ranges**

Anomaly

Timestamp

**Predicted Anomaly Ranges**

Anomaly

Timestamp

# Real Anomaly Ranges

# Predicted Anomaly Ranges

Sine, Luminol

**Real Anomaly Ranges**

**Predicted Anomaly Ranges**

## Real Anomaly Ranges

Anomaly

Timestamp

## Predicted Anomaly Ranges

Anomaly

Timestamp

ECG, Greenhouse

Real Anomaly Ranges

Predicted Anomaly Ranges

ECG, Luminol

**Real Anomaly Ranges**

Anomaly

Timestamp

**Predicted Anomaly Ranges**

Anomaly

Timestamp

NYC-Taxi, LSTM-AD

# NYC-Taxi, Greenhouse

## Real Anomaly Ranges

## Predicted Anomaly Ranges

NYC-Taxi, Luminol

**Real Anomaly Ranges**

**Predicted Anomaly Ranges**

[Supplementary Material 2 · Paper968-supplementary-SingleDetector.pdf]

# NeurIPS'18 Paper #968:
# Precision and Recall for Time Series

Supplementary Material

for Sections 5.2 and 5.3

# Readme

- This file contains line graphs for our 7 experimental datasets for LSTM-AD.

- For each dataset, we plot real and predicted anomaly ranges for their respective time series.

- On each graph, x-axis represents time and y-axis denotes where anomalies are. (t,1) means that the data value at timestamp x=t is anomalous, whereas (t,0) means that it is not.

- We used these graphs as reference in understanding and explaining the experimental results in Sections 5.2 and 5.3 of our paper.

## Real Anomaly Ranges

## Predicted Anomaly Ranges

# Space-Shuttle, LSTM-AD

Sine, LSTM-AD

**Real Anomaly Ranges**

**Predicted Anomaly Ranges**

## Real Anomaly Ranges

## Predicted Anomaly Ranges

## Real Anomaly Ranges

## Predicted Anomaly Ranges

Machine-Temp, LSTM-AD

**Real Anomaly Ranges**

**Predicted Anomaly Ranges**