[Reviews · NeurIPS 2018]

Reviewer 1



In this paper the authors define a new set of metrics for evaluating the performance of time series anomaly detection algorithms. They argue (and demonstrate) that existing precision and recall metrics that have been designed for point anomaly detection algorithm evaluation, do a poor job of estimating the quality of results for time series anomalies. This is actually a very important problem in the domain of time series anomaly detection, and has not been addressed in the literature, except in very specific context, such as the Numenta Scoring Model. The authors pay attention to important aspects of sequence anomaly detection such as existence (the only one previously considered), overlap, and position bias in coming up with a final score. They show how under certain assumptions, these definitions align with the point based metrics. The fact that these proposed metrics are tunable make it suitable for any anomaly detection task, unlike the Numenta model. Strengths: 1. Paper is well written and easy to understand. Notations are clearly defined. 2. The metrics are well thought out and appear to cover all aspects of time series anomaly detection. 3. Experimental results demonstrate the superiority of this evaluation framework over existing models. Weaknesses: 1. Figures (especially Figure 3) are hard to read and should be made bigger 2. Although this paper is not about comparing different anomaly detection frameworks, the paper would have been more complete if it compared results from at least one more detection technique other than the LSTM-AD approach to show that the evaluation framework produces similar results across multiple anomaly detection techniques for the same data set. This paper can open up the basis for designing new anomaly detection algorithms using these metrics for optimization. Updated after author responses: I am happy with the reviewer responses and do not plan to change my decision.

Reviewer 2



Summary: The authors present a parameterized model that generalizes precision and recall to "range-based" anomaly detection. A range-based anomaly is defined as "an anomaly that occurs over a consecutive sequence of time points, where no non-anomalous data points exist between the beginning and the end of the anomaly". The model is carefully and precisely defined. Then it is evaluated by comparing it against the metric proposed in the Numenta Anomaly Benchmark. The authors conclude that their metric is superior to the NAB metric. Assessment: I like this paper, because it reflects clear thinking about problem formulation that will be valuable to people working in this area. Clarity: The paper is well-written. I had a few questions. At line 269, what does "recessing" mean? In general, the paper would be improved if the authors provided better justification for biases beyond the front-end bias. I didn't see how their examples related to mid- and back-end bias. I can't imagine a case in which one would want a back-end bias. You need to persuade the reader that a problem exists before providing a solution. I think you have an error at line 22. In the context of anomaly detection, recall is the fraction of true anomalies that were detected and precision is the fraction of detections that were true anomalies. This is how you define things later, but the definitions here seem focused on detecting non-anomalies. Technical Soundness: The main claim of the paper is that their model can be used to "evaluate, rank, and compare results of AD algorithms". However, their model does not give a single measure but instead amounts to a domain-specific language for expressing various notions of precision and recall. While any specific instantiation of the model can be applied to evaluate, rank, and compare AD algorithms, it isn't clear how to do this in general. Indeed, is that a realistic goal? Every anomaly detection application has its unique costs and interaction protocol. For example, if we consider fraud detection, we are typically interested in the precision @ K: how many of the K top-ranked alarms are true alarms? This is because our scarce resource is the fraud analyst, and he/she can only examine K candidates per day. The measure described in this paper could give us a good definition of precision, but I would want to then rank by precision @ K. I might also want to rank by the financial cost or other features of the anomaly. These are application-specific measures that I don't think can be expressed by the current model (except for the simple measure of the size (duration) of the anomaly). If we consider equipment diagnosis, the natural measure is precision @ 100% recall (or some high percentile such as 95%). Again, the definitions in this paper would correctly define precision and recall, but I would then compare algorithms based on a measure computed from those. It is hard to assess the comparison with the NAB scoring model. Are F1, F2, and F0.5 the right measures? They make sense for information retrieval, but in my experience, they rarely make sense for anomaly detection applications. A stronger paper would have performed a series of case studies where, in each case, the application-specific criterion would have been expressed in the model. I would suggest replacing section 6.4 with this kind of analysis. To me, it is obvious that the metrics could be computed efficiently, but I understand that other readers might want some reassurance on this point. (Minor note: You could express the asymptotic complexity as O(max{N_r,N_p}).) I thought the experiment reported in Table 2 was very nice. Originality: I believe the work is original. Significance: The paper reflects clear thinking about the problem which will be valuable to anyone who works in this area.

Reviewer 3



The paper discusses a new approach to evaluating range based anomaly detection algorithms. It provides unable recall and precision evaluation metrics. 1. quality Overall the paper is well written and a good discussions on precision and recall philosophy and why the need to update the evaluation method for range based anomalies. The range based recall discussion is good. Balancing between existence and overlap is welcome but also introduces a number of tunable parameters that raise some questions on how best these would be setup for different use cases. This increases the complexity of comparisons between methods. - How will learning algorithms be tuned given the different evaluation tuning parameters. Other methods aren't optimising for this (including LSTM AD)? - What happens when during evaluation when the anomaly distribution changes, necessitating different tuning parameters for evaluation? Would have been great to also have ARIMA, or simpler time series anomaly detection model to start hinting on how such models would be augmented to be able to best fit the metrics. Small typo, page 6 line 263. Should be single-point 2. clarity Overall the writing is clear. I however, even if it is for space reasons, feels that both figure 4 and figure 5 are not readable on printed paper. I had to use the pdf and zoom on. There should be a limit on how small the font should be. This is a challenge because you refer to the figures for the comparisons in the experimental section. The development of recall and precision ranged based metrics are well explained. 3. originality This is a good tun-able evaluation method that takes into account the challenges of limiting the scope of what precision and recall are in the case or ranged based anomalies. The work shows why ranged based anomaly evaluation is important and how the typical recall and precision measures are a special case of the ranged based evaluation methods developed. Further benchmarking and comparisons against other anomaly benchmarks are shown and a discussion on limitations both of this evaluation model and the other benchmarks discussed. 4. and significance The paper does address an important challenge in the anomaly detection domain. Dealing with non-point based anomalies is important. Think of doing conditioning monitoring of different processes. It is not sufficient in all cases to identify that there was an anomaly, but being able to find how long it lasted for and exactly which period will assist heavily with troubleshooting. This models provides an approach that can be used to capture such requirements while allowing flexibility to reduce to a classical recall or precision evaluation. The experiments cover a number of properties that are important and highlight properties of both the developed evaluation measures and the benchmarks metrics. I appreciate the discussion on the computational cost of finding the anomalies and how there could be some optimisations. - How will this impact the performance in a streaming anomaly detection setting if one has to use the middle and flat anomaly detection?